# Antimicrobial Activities of α-Helix and *β*-Sheet Peptides against the Major Bovine Respiratory Disease Agent, *Mannheimia haemolytica*

**DOI:** 10.3390/ijms25084164

**Published:** 2024-04-09

**Authors:** Ruina Bao, Zhi Ma, Kim Stanford, Tim A. McAllister, Yan D. Niu

**Affiliations:** 1Faculty of Veterinary Medicine, University of Calgary, Calgary, AB T2N 4Z6, Canada; ruina.bao@ucalgary.ca; 2College of Biotechnology and Bioengineering, Zhejiang University of Technology, Hangzhou 310014, China; mazhi@zjut.edu.cn; 3Department of Biological Sciences, University of Lethbridge, Lethbridge, AB T1K 3M4, Canada; kim.stanford@uleth.ca; 4Agriculture and Agri-Food Canada, Lethbridge Research and Development Center, Lethbridge, AB T1J 4B1, Canada; tim.mcallister@agr.gc.ca

**Keywords:** bovine respiratory disease, antimicrobial peptide, *Mannheimia haemolytica*, antibacterial activity

## Abstract

Bovine respiratory disease (BRD) is the leading cause of morbidity and mortality in cattle raised in North America. At the feedlot, cattle are subject to metaphylactic treatment with macrolides to prevent BRD, a practice that may promote antimicrobial resistance and has resulted in an urgent need for novel strategies. *Mannheimia haemolytica* is one of the major bacterial agents of BRD. The inhibitory effects of two amphipathic, α-helical (PRW4, WRL3) and one β-sheet (WK2) antimicrobial peptides were evaluated against multidrug-resistant (MDR) *M. haemolytica* isolated from Alberta feedlots. WK2 was not cytotoxic against bovine turbinate (BT) cells by the MTT (3-(4,5-dimethylthiazol-2-yl)-2,5-diphenyltetrazolium bromide) assay. All three peptides inhibited *M. haemolytica*, with WK2 being the most efficacious against multiple isolates. At 8–16 µg/mL, WK2 was bactericidal against Mh 330 in broth, and at 32 µg/mL in the presence of BT cells, it reduced the population by 3 logs CFU/mL without causing cytotoxic effects. The membrane integrity of Mh 330 was examined using NPN (1-N-phenylnaphthylamine) and ONPG (o-Nitrophenyl β-D-galactopyranoside), with both the inner and outer membranes being compromised. Thus, WK2 may be a viable alternative to the use of macrolides as part of BRD prevention and treatment strategies.

## 1. Introduction

*Mannheimia haemolytica* is an opportunistic pathogen and a major bacterial agent of bovine respiratory disease (BRD)—a leading cause of morbidity and mortality in cattle raised in North American feedlots [1,2]. The bacterium has been responsible for fibrinous pleuropneumonia in ruminants [2]. Commensal *M. haemolytica* are found in the nasopharynx, upper respiratory tract, and tonsillar crypts of healthy cattle. Isolation of *M. haemolytica* from North American cattle with BRD identified serotype 1 (44%), serotype 2 (30%), and serotype 6 (17%) as the predominant serotypes [3]. Presently, cattle typically receive metaphylactic macrolides upon arrival at the feedlot in an attempt to reduce the incidence of BRD [4,5]. This practice may contribute to an increase in the prevalence of multidrug-resistant (MDR) *M. haemolytica* and *P. multocida* [1,6]. Vaccines are of variable efficacy, although they could possibly reduce the use of antimicrobials [4,7,8]. However, a meta-analysis of 14 vaccine studies showed no significant difference in incidence of BRD in vaccinated cattle within the first 45 days of residency within feedlots [4]. Another study of a pentavalent modified-live virus respiratory vaccine showed no significant difference in BRD morbidity of vaccinated vs. unvaccinated control cattle, but metaphylaxis with tulathromycin did decrease the incidence of BRD [7]. In contrast, a recent study evaluating the efficacy of administering a modified-live virus vaccine against infectious bovine rhinotracheitis, bovine virus diarrhea virus Type 1 and 2, bovine respiratory syncytial virus, and parainfluenza Type 3 virus found a reduction in chronically ill cattle (i.e., cattle experiencing more than three instances of BRD) and a reduction in antimicrobial use in cattle vaccinated on day 1 versus on day 28 [8].

Antimicrobial peptides (AMPs) are produced by the natural host innate immune system [9]. Cattle produce bovine cathelicidins with antimicrobial activity against human pathogens [10]. Bacteria also produce AMPs, called bacteriocins, that provide them with a competitive advantage in acquiring nutrients and establishing in biological niches [9]. Antimicrobial peptides are a feasible alternative to antibiotics due to their multifunctional properties in addition to their broad-spectrum antimicrobial activity. For example, AMPs modulate intestinal microbiota, promote nutrient absorption, reduce greenhouse gas emission, and promote growth in broilers [11]. AMPs disrupt bacterial membranes or target intracellular components that are required for functional biochemical pathways [12,13]. Researchers have drawn upon AMPs of diverse origin (from bacteriocins to human defensins), but in general they share common biological functions such as membrane disruption and host immunomodulatory activity [12].

Many AMPs have already been approved for clinical use. Nisin A is a food preservative, and polymyxins B and E are prescribed for the treatment of eye and wound infections [13]. In livestock production, a combination of nisin and lysostaphin eradicated *Staphylococcus aureus* biofilms in dairy cows with bovine mastitis. Human β-Defensin-3 peptide analogue was studied for its effects against *M. haemolytica* [14]. Previous studies have also examined the antimicrobial activity of peptides derived from bovine NK-lysins against *M. haemolytica*, *P. multocida*, and *H. somni* [15]. However, these studies did not investigate the inhibitory activity of the peptides in the presence of serum or saline, conditions that would be more reflective of the bovine lung. Although bovine tracheal antimicrobial peptide (TAP) exhibited high activity (1.56 µg/mL) against *M. haemolytica* in vitro, it failed to control this pathogen in a calf challenge model [16]. Later studies showed that the activity of TAP was attenuated in the presence of >25 mM of sodium chloride and >1.25% serum [16].

Previously, three synthetic peptides, PRW4, WRL3, and WK2, inhibited methicillin-resistant *S. aureus* strains, *Escherichia coli* and *Candida albicans* isolated from humans [17,18,19]. Design of PRW4 (RFRRLRWKTRWRLK KI-NH2), an α-helical peptide, was based on the amphipathic portion of porcine antimicrobial peptide PMAP-36 [17]. WRL3, an α-helical peptide (WLRAFRRLVRRLARGLRR-NH2), and WK2 (WKWKCTKSGCKWKW-NH2), a β-sheet peptide, were derived from bacteriocin Leucocin A, produced by *Leuconostoc gelidum* [17,18]. The objectives of this study were to evaluate the antimicrobial activity of PRW4, WRL3, and WK2 against *M. haemolytica* isolated from Alberta feedlots under both planktonic conditions and when cultured with bovine turbinate cells. An overview of the experimental plan is shown in Figure 1. Promising AMPs with superior efficacy were further assessed for cytotoxicity and their ability to promote membrane permeability in *M. haemolytica*.

## 2. Results

Inhibitory activity of the three synthesized peptides (Table 1) varied among *M. haemolytica* strains (Table 2), with WRL3 and WK2 being more effective (minimum inhibitory concentration (MIC) 2 to 32 µg/mL) than PRW4 (MIC ~64→256 µg/mL (Table 3)). Excluding the activity of WRL3 against Mh 587 and WK2 against Mh 13, bacteria were generally not recoverable from the MIC well. Previously, the MICs of WRL3 and WK2 against a panel of pathogens such as STEC and S. enterica serovar *Typhimurium* were found to be 8–32 µg/mL [17] and 2–4 µg/mL [18].

Viability of BTs cells did not differ (*p* > 0.05) in comparison to media control when WK2 was added at concentrations of 8–128 µg/mL (Figure 2A). However, with WRL3, viability of BTs cells decreased (*p* < 0.05) based on a decline in mitochondrial dehydrogenase activity at concentrations higher than 4 µg/mL (Figure 2B).

WK2 inhibited *M. haemolytica* Mh 330 when diluted in MHB (Figure 3A), saline (Figure 3B), or serum (Figure 3C). Interestingly, Mh 330 grew better in media supplemented with 2.5% FBS; however, despite the higher cellular density, the MIC in the presence of FBS remained at 8 µg/mL.

*M. haemolytica* isolates were screened for their ability to form biofilms on polystyrene plates to select the strongest biofilm former. The crystal violet assay quantified the biomass after 48 h incubation at 37 °C, with 5% CO_2_ in RPMI media. Mh 330 consistently produced the most robust biofilm mass on polystyrene, defined as an OD reading of more than four times the optical cut-off value (Figure 4A). This strain also readily adhered to BTs (Figure 4B). After four washes, the adherent population of Mh 330 remained ~10^6^ CFU/mL, based on counts after BTs were lysed. There was no difference (*p* > 0.05) between the number of bacteria attached to tissue-culture treated polystyrene and BTs.

In comparison to the buffer and bacteria-only controls, *M. haemolytica* Mh 330 treated with WK2 at 1× and 2× MIC (8–16 µg/mL) had greater fluorescence, which was indicative of greater outer membrane damage (*p* < 0.001, Figure 5A). When ONPG is cleaved by intracellular β-galactosidase, the colorless compound is separated into D-galactose and ortho-nitrophenol, producing a yellow color [10]. WK2 was able to increase the inner membrane permeability of *M. haemolytica* Mh 330 after 15 min of exposure at the 1× MIC but no incremental increase (*p* > 0.05) was observed at 2× the MIC (Figure 5B).

WK2 inhibited the growth of Mh 330 bacterial colonies in the presence of BTs. Only the 4× MIC (32 µg/mL) was able to considerably reduce bacterial concentration by 3 logs CFU/mL (*p* < 0.05) (Figure 6). There was no inhibitory effect on *M. haemolytica* Mh 330 from the peptide diluent alone.

## 3. Discussion

This study is the first systematic in vitro evaluation of the utility of synthetic α-helix and β-sheet peptides for therapeutic application against MDR *M. haemolytica*. WRL3 (α-helix) and WK2 (β-sheet) were capable of killing *M. haemolytica*, with the exception of Mh 13 at the concentration of 4–32 µg/mL, in comparison to PRW4, which failed to exert an antibacterial effect against serotype 1 strains of *M. haemolytica* (MIC > 256 µg/mL). Notably, WK2 demonstrated superior anti-*M. haemolytica* activity by exhibiting tolerance against physiological concentrations of NaCl or 2.5% fetal bovine serum and by reducing the pathogen population in the presence of BTs. In contrast to WRL3, WK2 was potent against *M. haemolytica* without cytotoxic effects. Moreover, the bactericidal activity of WK2 was exerted by permeabilization of both inner and outer membranes of the bacterial pathogen. The relatively higher MIC of PRW4 against *M. haemolytica* may be attributed to its grand average of hydropathy (GRAVY) score of −1.644 in comparison to −1.429 for WK2 (calculated from ExPASy based on the method proposed previously [20]), suggesting that PRW4 is less hydrophobic than WK2. There is evidence correlating greater membrane lysis activity to increasing peptide hydrophobicity [21,22]. WK2 is also the most stable peptide, with the highest predicted in vivo half-life according to the instability index as determined by ExPASy.

Previously, PRW4 was shown to inhibit *E. coli* strains, and *S. Typhimurium* 7731 at 9.2 µg/mL and 8 µM 18.4 µg/mL. However, it only inhibited *M. haemolytica* Mh 587 at concentrations of 64 µg/mL and Mh 276 at 256 µg/mL. Mh 587 was consistently the most susceptible isolate to all three peptides (Table 2). The strain is serotype 2, which is considered a commensal resident in the respiratory tract of cattle. Isolate Mh 587 was sampled from the nasopharynx of a healthy animal in Alberta—with no integrative conjugative elements (ICE) found [23]. In contrast, isolate Mh 13 was obtained from lung tissue of a BRD mortality [24]. Its genomic sequence contained five ICE-related genes (*int2*, *rel1*, *mco*, *tnpA*, and *parB*) and multidrug-resistant genes [24]. However, this result is surprising given that researchers have found collateral sensitivity to antimicrobial peptides in antibiotic-resistant bacteria [25]. Thus, further studies are required to determine what genetic components contribute to the high WK2 minimum bactericidal concentration for Mh 13.

WRL3 exerted strong antibacterial effects (4 µg/mL to 32 µg/mL) against a wide range of Gram-negative and -positive bacteria and yeast, including *E. coli* O157: H7 21530, *S. paratyphi*, *S. typhimurium* 51005, methicillin-resistant *Staphylococcus aureus* (MRSA) 43300, and *Candida albicans* 10231 [26]. Furthermore, it was capable of reducing populations of MRSA in a mouse model of infected skin wound alone and in combination with ceftriaxone [26]. In this study, WK2 was bactericidal against MDR *M. haemolytica* strains at a similar range of concentrations (8 to 32 µg/mL) that was effective against the other Gram-negative pathogens. In comparison to PRW4 and WK2, WRL3 was bactericidal against all *M. haemolytica* strains, including Mh 13. However, WRL3, which had previously exhibited cell-selective killing ability of MRSA when bacteria were cocultured with erythrocytes and macrophages from murine line RAW264.7 [26], it was cytotoxic to BTs at concentrations above 4 μg/mL. In contrast, WK2 was not harmful towards BTs even at 128 μg/mL which has clinical implications regarding the amount of antimicrobial peptide that may be safely administered in cattle. Although this model may be used as a high-throughput method of assessing efficacy of antimicrobial peptides, animal studies are required to examine their half-life in vivo and systemic toxicity.

WK2 was examined further in the salt and serum sensitivity assays. Previously Vulikh and colleagues [16] found that tracheal antimicrobial peptide (TAP), a β-defensin of 38 amino acid residues in length, showed inhibitory activity against *M. haemolytica* in vitro but was nonprotective in 1-month-old calves challenged with *M. haemolytica*. TAP is produced naturally by bovine airway epithelial cells. The researchers found that the presence of physiological NaCl (150 mM) and 2.5% bovine serum inhibited TAP antibacterial activity in MIC assays. However, unlike TAP, the antimicrobial potency of WK2 was not sensitive to physiological salt and serum, which corroborates with previous findings that WK2 is effective in a mouse model of *S. typhimurium* infection [18].

Prior experiments demonstrated that WK2 was effective against *S. typhimurium* DT104 at 4 μg/mL [18]. Similarly, it was bacteriostatic at the same range (8–32 μg/mL) against all strains of *M. haemolytica* studied. WK2 has a charge of +6 while WRL3 has a charge of +9. Previously, it was shown that an increase in charge beyond +5 correlated with an increase in cytotoxicity [27]. The optimal charge of WK2 may contribute to its cell-selective killing and low cytotoxicity. However, the inability of WK2 to kill Mh 13 at 128 µg/mL, >3× MIC, warrants further experiments into nucleotide polymorphisms between this isolate and the other susceptible serotype 1 strains, which may contribute to their susceptibility to AMP. Genes of interest imparting resistance to antimicrobial peptides include the PhoP/PhoQ system. For example, in *Salmonella*, PhoP-activated genes (*pag* genes) and PhoP repressed genes (*prg* genes) are essential to aminoarabinose modification of lipid A in Gram-negative outer membranes, a target of polymyxin B [28]. The mechanism of action of polymyxin B is similar to antimicrobial peptides in that it interacts with both the outer and cytoplasmic membranes [29].

In comparison to the previous papers published on the mechanism of action of WK2 on *Salmonella*, the AMP exerted the same bactericidal mechanism against *M. haemolytica* via outer membrane and inner membrane permeability at the MIC. AMPs typically disrupt the bacterial membranes, leading to cellular leakage [30]. Possible future experiments to perform include examining whether WK2 also has intracellular targets against *M. haemolytica* strains such as nucleic acid binding and antibiofilm properties via downregulation of fimbriae similar to WK2’s effects on *Salmonella* [18,19].

Evidence in the literature shows that BRD bacterial agents *M. haemolytica*, *Histophilus somni*, and *P. multocida* can form biofilms [31,32]. Colonies within a surface-attached microbial community are more drug-recalcitrant than planktonic bacteria—for *M. haemolytica*, sessile bacteria were less sensitive to erythromycin, ampicillin, and trimethoprim/sulfadoxine [33]. Cattle experiencing chronic BRD may have multispecies biofilm formation in their lower respiratory pathway [30,34]. In a scanning electron micrograph, *M. haemolytica* was also observed to exist in microcolonies in the lungs of cattle with BRD [35]. An unexpected result from the biofilm assays was the rarity of biofilm formation within the collection of *M. haemolytica* isolates. Previously, it was shown that approximately 10% of the inoculum containing a serotype 2 strain of *M. haemolytica* adhered to 21-day old, differentiated, primary bovine bronchial epithelial cells grown in air liquid medium at 24 h, while serotype 1 strain readily colonized and proliferated (>1000% inoculum) within the differentiated bovine bronchial epithelial cell cultures [35]. Among the isolates studied, Mh 330 was determined to be the most robust biofilm former on polystyrene based on its biomass quantified by the crystal violet assay and bacteria colonies recovered. In the presence of BTs, WK2 reduced the number of bacteria recovered by 3 logs CFU/mL at a concentration of 32 µg/mL. Based on observations under the microscope after the three-hour incubation period and plating 100 μL of the WK2-treated wells onto 5% sheep blood agar, WK2 was able to reduce the recovered population of Mh 330 selectively without disturbing the BT monolayer. One research group proposed administering six *Lactobacillus* strains in the form of an intranasal vaccine to prevent adhesion of pathogenic *M. haemolytica* and colonization in the nasal passage [36,37]. Additional studies on the presence/absence of systemic toxic effects and in vivo bioavailability of WK2 are required to demonstrate the efficacy of delivering WK2 intranasally to prevent bacterial pathogens from colonizing the nasopharynx and the lungs of healthy cattle.

In the study, the absence of mucus and use of monoculture was not able to recapitulate the complexity of interactions between pathogenic *M. haemolytica*, microbiome, and host immune response in the bovine respiratory tract, which may impact in vivo antimicrobial activity of the peptides. Further evaluation of the AMPs against other BRD bacterial pathogens (e.g., *P. multocida* and *H. somni*) and their potential immunomodulatory effects for respiratory infection is warranted because previous research showed multispecies biofilm formation in the lower respiratory tract of cattle [34].

## 4. Materials and Methods

### 4.1. Synthesis of the Peptides

Antimicrobial peptides PRW4, WRL3, and WK2 were synthesized and purified by Canada Peptide (Pointe-Claire, QC, Canada). The peptide sequences synthesized in this study are listed in Table 1. The peptides were determined to be >95.0% pure based on high-performance liquid chromatography. The lyophilized peptides were aliquoted in 1 mg fractions in screw cap vials and stored at 4 °C. Before use, peptides were reconstituted in sterile ultrapure water at a concentration of 512 μg/mL.

### 4.2. Bacterial Strains and Cell Culture

The bacterial strains *E. coli* ATCC 25922, Shiga-toxin producing *E. coli* (STEC) O91 EC20010076, Mh 276, Mh 13, Mh 136, Mh 535, Mh 587, and Mh 330 were used in the study (Table 2 [23,24,38]). Glycerol stocks of isolates were streaked onto tryptic soy agar (TSA; EMD Millipore Corporation, Billerica, MA, USA) supplemented with 5% defibrinated sheep blood (QUAD FIVE, Ryegate, MT, USA) and incubated at 37 °C in the presence of 5% CO_2_ for 20–24 h. A single colony was inoculated into 10 mL Mueller Hinton broth (MHB; Oxoid, Basingstoke, Hants, UK) and incubated at 37 °C with shaking at 180 rpm for 20–22 h. Bovine turbinate cells (BTs) were grown on Dulbecco’s Modified Eagle’s Media (DMEM) (BioWhittaker, Walkersville, MD, USA) with phenol red, 10% fetal bovine serum (Gibco by Life Technologies, Grand Island, NY, USA), and 1% Antibiotic-Antimycotic (Gibco, Life Technologies, Grand Island, NY, USA) at 37 °C with 5% CO_2_.

### 4.3. In Vitro Minimum Inhibitory Concentration (MIC) Assays of M. haemolytica

The microbroth dilution method previously described [26] was used to determine the MIC. To prevent peptides from adhering to plastic wells, peptides were dissolved in an aqueous solution consisting of 0.01% glacial acetic acid (EMD Millipore) and 0.2% bovine serum albumin (BSA; Cytiva, Logan, Utah, USA), referred to as peptide diluent in subsequent methods. Early log phase *M. haemolytica* cultures were prepared by adding 500 μL of an overnight culture into 5 mL MHB and grown in a shaking incubator (180 rpm) at 37 °C for 2.5 h, at which point an optical density (OD_600nm_) of ~0.2 was reached. Bacterial suspensions (50 μL) containing 10^5^ CFU/mL of early log phase *M. haemolytica* were inoculated into 96-well polypropylene plate (Corning Incorporated, Kennebunk, ME, USA) containing peptide solution at concentrations containing 256 µg/mL, 128 µg/mL, 64 µg/mL, 32 µg/mL, 16 µg/mL, 8 µg/mL, 4 µg/mL, 2 µg/mL, 1 µg/mL, 0.5 µg/mL, and 0.25 µg/mL). See Appendix A plate layout for determining the minimum inhibitory concentration of antimicrobial peptides against *M. haemolytica*.

The MIC was determined as the minimal concentration that completely inhibited bacterial growth based on lack of visual turbidity [26]. Polymyxin B (EMD Millipore Corp., Copenhagen, Denmark) and Ceftiofur (Sigma Aldrich, St. Louis, MO, USA) were used as positive controls. Wells containing bacteria only with peptide diluent (50 μL of 10^5^ CFU/mL) served as negative controls. The MICs of PRW4, WRL3, and WK2 were conducted in at least 2 independent trials with two technical replicates within each of the runs. Absence or occurrence of growth to estimate the minimum bactericidal concentration (MBC) was determined by plating the first three wells that lacked turbidity on TSA with 5% sheep blood. Plates were incubated for 20–24 h at 37 °C with 5% CO_2_ and subsequently examined for the presence of colonies.

### 4.4. Assess Effect of Salt and Serum on Antibacterial Activity of Antimicrobial Peptides

To assess whether the antibacterial activity of the peptides was affected by physiological sodium chloride and serum, 150 mM of NaCl was dissolved in MHB and 2.5% fetal bovine serum was added to MHB after autoclaving. The concentration used for NaCl (150 mM) was referenced from a previous paper on WRL3, which examined NaCl amongst other salts on its antibacterial activity [31]. The concentration of fetal bovine serum (2.5%) was referenced from a study where the presence of 2.5% serum attenuated the antibacterial effect of tracheal antimicrobial peptide [16].

### 4.5. Assess Cytotoxic Effects Using the MTT (3-(4,5-dimethylthiozol-2-yl)-2,5-diphenyltetrazolium Bromide) Assay

The cytotoxicity of the peptides against BTs were assessed using the MTT 3-(4,5-dimethylthiozol-2-yl)-2,5-diphenyltetrazolium bromide assay, performed according to the CyQUANT Cell proliferation assay kit instructions (Life Technologies Corporation, Eugene, OR, USA). In this assay, mitochondrial dehydrogenase from metabolically active cells reduces light-yellow/colorless MTT into insoluble, purple formazan that is dissolved by dimethyl sulfoxide (DMSO). The cytotoxicity of the peptides was compared to positive control wells containing 100 μL Dulbecco’s Modified Eagle Medium (DMEM) only applied to bovine turbinates. A stock solution of MTT (5 mg/mL) was dissolved in sterile phosphate-buffered saline (Fisher Scientific, Ottawa, ON, Canada) (pH = 7.4) and stored in the dark at 4 °C. Bovine turbinate cells (10^5^) in 100 μL of DMEM were seeded into 96-well polystyrene culture plates (Fisher Scientific, Rochester, NY, USA) and grown to 100% confluency at 37 °C with 5% CO_2_. The confluency and integrity of the BT monolayers were checked under the microscope (Nikon Eclipse TS100, Nikon, Tokyo, Japan). Before the start of the assay, exhausted media were removed by a pipette and replaced with 100 μL of freshly prepared peptides (ranging from 128 μg to 4 μg/mL) dissolved in DMEM with no phenol red (Fisher Scientific, Grand Island, NY, USA) to avoid a high background reading. Next, MTT stock solution (10 μL) was added to each well containing BT cells and peptide solutions and incubated at 37 °C for 4 h. After incubation, 85 μL from each well was removed and 50 μL of DMSO was added to each well to dissolve the reduced form of MTT (formazan). Plates were incubated for an additional 10 min at 37 °C before reading at OD_540 nm_ on BioTek Epoch 2 microplate spectrophotometer (Agilent Technologies, Santa Clara, CA, USA). Biological replicates refer to BTs seeded into 96-well plates on three separate occasions. Technical replicates refer to number of wells per peptide concentration performed each day. Three biological replicates with three technical replicates were performed for each peptide dilution. Three biological replicates with at least two technical replicates were performed for wells containing BTs with DMEM only. A single peptide was tested in a single 96-well plate.

### 4.6. Biofilm Screening

The biofilm formation assay was conducted as previously described [36]. *M. haemolytica* isolates were cultured in brain heart infusion broth (BHI) at 37 °C with shaking at 180 rpm for 20–22 h. Overnight cultures (~5 × 10^8^ CFU/mL) were diluted to 10^6^ CFU/mL in RPMI-1640 without L-glutamine (Corning Mediatech, Manassas, VA, USA) supplemented with 2 g/L sodium bicarbonate (Sigma-Aldrich, St. Louis, MO, USA). Diluted overnight culture (200 μL) was deposited in polystyrene 96-well plates (Fisher Scientific, Rochester, NY, USA) and incubated at 37 °C for 48 h under 5% CO_2_. After incubation, wells were washed twice with 250 μL ultrapure water and air-dried for 20 min. Crystal violet dye (Sigma-Aldrich, St. Louis, MO, USA) was added to each well and incubated at room temperature for 20 min. The dye was removed by a pipette, the wells were washed twice with 250 μL of sterile ultrapure water, and the plate was air-dried for 20 min. Stained biofilms were solubilized with 33% glacial acetic acid (EMD Millipore Corporation, Billerica, MA, USA), and absorbance was read at OD_595 nm_. Three biological replicates from three different colonies used to grow cultures and eight technical replicates per culture were conducted to quantify biofilm formation of Mh 330. The sum of the average of the negative control wells containing growth medium and its standard deviation multiplied by three was used as the optical cut-off value to indicate the background reading from the assay. Strains were classified based on the following: OD_595 nm_ reading ≤ ODcutoff, no biofilm producer; ODcutoff < OD_595 nm_ ≤ 2× ODcutoff, weak biofilm producer; 2× ODcutoff < OD_595 nm_ ≤ 4× ODcutoff, intermediate biofilm producer; and 4× ODcutoff < OD_595 nm_, strong biofilm producer [38]. STEC O91 EC20010076 has been previously identified as a strong biofilm former and was used as the positive control [38].

### 4.7. Attachment of M. haemolytica to Bovine Turbinates

Prior to the start of the experiment, the BTs were washed twice in antibiotic and serum-free DMEM and incubated for 1 h prior to inoculation. As the strongest biofilm former, Mh 330 was used to perform attachment assays to BT using the assay of [39] with minor modifications. The BTs were seeded into a 24-well flat bottom tissue culture treated plate (Corning, Corning, NY, USA) at 10^5^ cells per well in 0.5 mL of DMEM. Bovine turbinate cells were maintained on DMEM, 10% fetal bovine serum, and 1% Antibiotic-Antimycotic at 37 °C with 5% CO_2_ until cells were 70% confluent. *M. haemolytica* Mh 330 overnight cultures were grown for 20–22 h in 10 mL BHI at 37 °C, 5% CO_2_, with shaking 150–180 rpm. The mixture was centrifuged at 4000 × *g* at 4 °C for 10 min and the pellet was resuspended in 1 mL of DMEM without antibiotic or serum to ~10^9^ CFU/mL. Cultures (50 μL, ~10^9^ CFU/mL) were inoculated into each well to achieve a multiplicity of infection of 1000:1 ~10^8^
*M. haemolytica* per 10^5^ BT cells in each well. The plate was incubated for 3 h at 37 °C with a 5% CO_2_ atmosphere. The monolayers were washed four times, with the supernatant and final wash fractions saved. To lyse the monolayers, 0.5 mL of 0.1% Triton X-100 (Bio Basic Canada, Markham, ON, Canada) diluted in phosphate-buffered saline (PBS), pH = 7.4, was added to each well and the plate was incubated for 30 min at room temperature with gentle shaking (150 rpm). The supernatant, final wash, and attached fractions were diluted in PBS and plated on TSA containing 5% sheep blood agar to enumerate *M. haemolytica* colonies.

### 4.8. Ability of WK2 to Prevent Attachment of M. haemolytica to Bovine Turbinates

Before inoculation, WK2 was added to each well containing confluent BTs in triplicate per concentration of the peptide at final concentrations of 1×, 2×, and 4× of the MIC determined from planktonic cultures (8 μg/mL), along with a media-only control. *M. haemolytica* and BT cells were inoculated in equal proportions (10^5^) into each well. Two biological replicates of seeded BT cells on two separate days and three technical replicates (seeded three wells) were performed for the DMEM-only group. Three biological replicates and three technical replicates were performed for treatments at 8 μg/mL, 16 μg/mL, 32 μg/mL, and peptide diluent: DMEM (50:50).

### 4.9. Membrane Permeability

The protocol for observing inner membrane permeability was conducted as described previously, with minor modifications [10]. *M. haemolytica* Mh 330 and *E. coli* ATCC 25922 were grown to mid-log phase culture (OD_600 nm_ ~ 0.4–0.5, equivalent to 3–4 × 10^8^ CFU/mL) in MHB without lactose. Cells were centrifuged at 4000× *g* for 10 min at room temperature and resuspended in sodium phosphate buffer (10 mM Na_2_HPO_4_, 10 mM NaH_2_PO_4_ and 100 mM NaCl (Fisher Scientific, Fair Lawn, NJ, USA), pH = 7.5), to an OD_600 nm_ ~ 0.5. Peptide solution (50 μL) was diluted to a final concentration of 1× MIC and 2× MIC in a 96-well polystyrene plate with sodium phosphate buffer as a negative control. Polymyxin B sulfate (EMD Millipore, Billerica, MA, USA) was used as the positive control. Mid-log phase culture (50 μL) was aliquoted into each well before 5 μL of stock *o*-Nitrophenyl β-D-galactopyranoside (ONPG) (Thermo Fisher Scientific, Rockford, IL, USA) solution (30 mM) dissolved in ultrapure water was added. Yellow color formation from β-galactosidase hydrolysis of ONPG into *o*-Nitrophenol (ONP) served as an indicator of membrane permeability and was quantified by reading OD_405_ after the plate was incubated at 37 °C for 15 min. Prior to conducting the ONPG experiment, the presence of β-galactosidase enzymatic activity was assessed by streaking *M. haemolytica* colonies on BHI agar both with or without 1 mM isopropyl β-D-1-thiogalactopyranoside (IPTG) and 20 mM of X-gal (5-bromo-4-chloro-3-indolyl β-D-galacto pyranoside). The resulting colonies were blue in color, suggesting that β-galactosidase was able to cleave X-gal (Appendix A). Peptide WK2 was assessed for its ability to rapidly permeabilize the inner membrane of *M. haemolytica* Mh 330 by observing the change in color after the addition of 1× and 2× MIC in comparison to the bacteria-only control. Three biological replicates (performed on three separate days using bacterial culture inoculated from single colonies) and at least four technical replicates were conducted for each experimental group.

Outer membrane permeability was assessed using the 1-N-phenylnapthylamine (NPN) fluorescent dye (Fisher Scientific, Fair Lawn, NJ, USA) as described previously, with minor modifications [40]. Mid-log phase cultures of Mh 330 and *E. coli* ATCC 25922 (OD_600 nm_ = 0.4–0.5, 4-5 × 10^8^ CFU/mL) were centrifuged at 4000× *g* for 10 min at room temperature to capture bacterial pellets. The pellets were resuspended in PBS (pH = 7.4) and 50 µL of the bacterial suspension was added to each well. In a black, clear-bottomed, polystyrene, 96-well plate (Eppendorf, Hamburg, Germany), 50 µL of final concentration 1× MIC WK2 was deposited and 50 µL of the mid-log inoculum was added to each well. A 5 mM stock solution of NPN in acetone (2 µL; VWR, Radnor, PA, USA) was added to each well. Using a fluorescent spectrophotometer (SpectraMax M2, Molecular Devices), the fluorescence values were read immediately, at OD_350 nm_ with an emission wavelength of OD_420 nm_. Outer membrane permeabilization was determined using the fluorescence dye NPN, where fluorescence is increased when the dye is in a hydrophobic environment, such as within the outer bacterial membrane [40]. Three biological replicates (we performed this experiment on three separate days using bacterial culture inoculated from single colonies) and at least four technical replicates were conducted for each experimental group.

### 4.10. Statistical Analysis

All statistical analysis was performed using GraphPad Prism version 9.1.1 for macOS, GraphPad Software, San Diego, CA, USA, www.graphpad.com. In experiments comparing bacteria colony-forming units, the y value (CFU/mL) was first log-transformed to reduce heteroscedasticity [20]. The differences in means between groups were compared using one-way analysis of variance (ANOVA) with the Tukey’s multiple comparisons’ post hoc test.

## 5. Conclusions

This study evaluated the potential of α-helix and β-sheet peptides for killing BRD bacterial pathogens. Heightened AMR is making treatment of BRD increasingly ineffective. The present study demonstrated that peptides are able to inhibit MDR *M. haemolytica* at biologically relevant conditions without cytotoxic effects, which suggests that AMPs may present a new avenue for effectively managing BRD in cattle. However, the bioavailability of AMPs, ease of degradation in vivo, the high costs of synthesis, and the presence/absence of systemic toxic effects are a few of the questions that researchers must consider before novel AMPs are ready for clinical use.

## Figures and Tables

**Figure 1 ijms-25-04164-f001:**
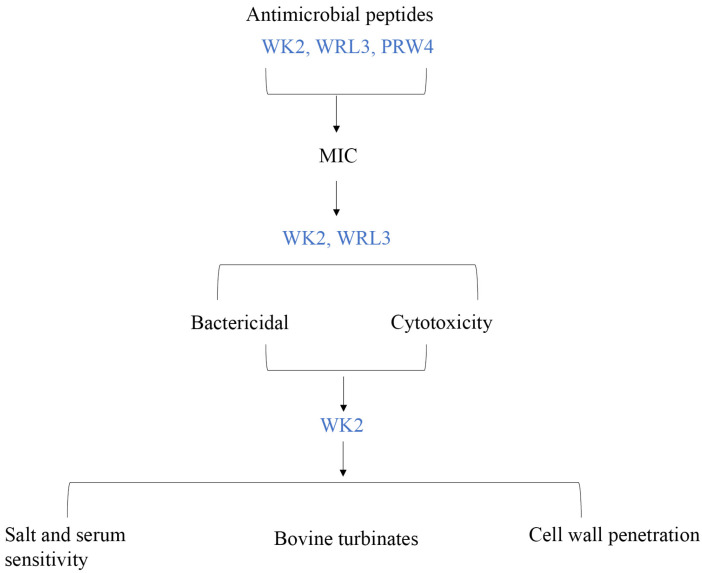
Experimental plan and antimicrobial peptides assessed at each stage.

**Figure 2 ijms-25-04164-f002:**
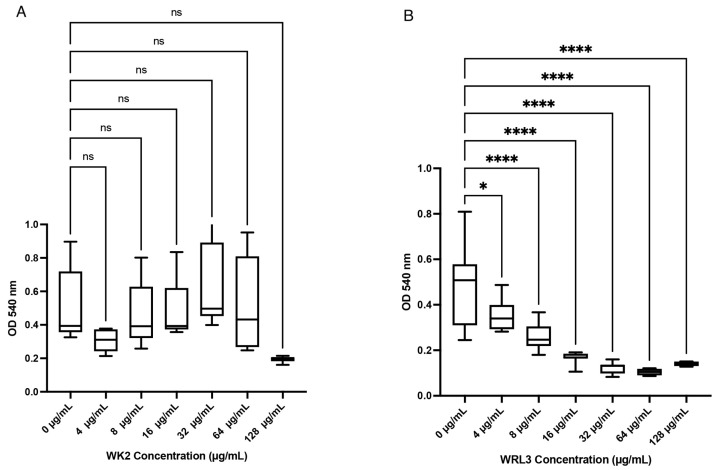
Cytotoxicity of antimicrobial peptides WK2 and WRL3 against BTs quantified by the MTT assay. (**A**) WK2 had no significant effects (ns: *p* > 0.05) on the mitochondrial dehydrogenase activity of BTs. (**B**) WRL3 at or above 4 µg/mL reduced BT cell viability (* *p* < 0.05, **** *p* < 0.0001). Statistical analysis was performed using one-way ANOVA with Tukey’s post hoc test. Results from three biological experiments performed in at least duplicates are presented as box and whisker graphs, where the line plotted in the middle is the median. Biological replicates refer to BT cells seeded into 96-well plates on different days. Technical replicates refer to number of wells per treatment performed each day.

**Figure 3 ijms-25-04164-f003:**
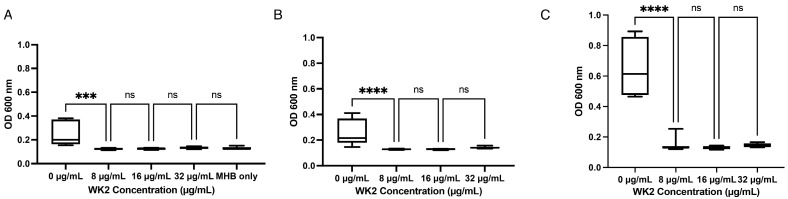
Salt and serum sensitivity of WK2 inhibitory activity against *M. haemolytica* Mh 330. (**A**) Inhibitory activity of WK2 in MHB (*** *p* < 0.001, ANOVA Tukey’s post hoc test). (**B**) Physiological concentration of NaCI (150 mM) does not affect WK2 antibacterial activity at 8 mg/mL (**** *p* < 0.0001, ANOVA Tukey’s post hoc test). (**C**) Presence of 2.5% fetal bovine serum does not affect WK2 antibacterial activity (ns: *p* > 0.05, ANOVA Tukey’s post hoc test). Results from three biological experiments performed in duplicates are presented as box and whisker graphs, where the line plotted in the middle is the median.

**Figure 4 ijms-25-04164-f004:**
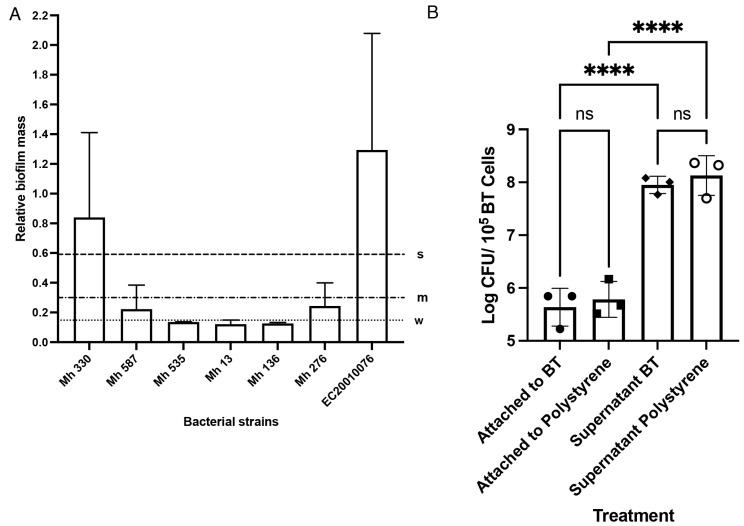
Mh 330 attaches strongly to abiotic and biotic surfaces. (**A**) Biofilm biomass formed by *M. haemolytica* strains quantified by crystal violet dye (s: strong, m: intermediate, w: weak). Data are presented as means and standard deviations of 3 experiments with 8 replicates. (**B**) Mh 330 exhibited equally strong attachment to tissue culture-treated polystyrene wells and BTs (ns: *p* > 0.05, ANOVA with Tukey’s post hoc test), while there were significantly more bacterial cells in the supernatant fraction (**** *p* < 0.0001, ANOVA with Tukey’s post hoc test, *n* = 3 performed in triplicate).

**Figure 5 ijms-25-04164-f005:**
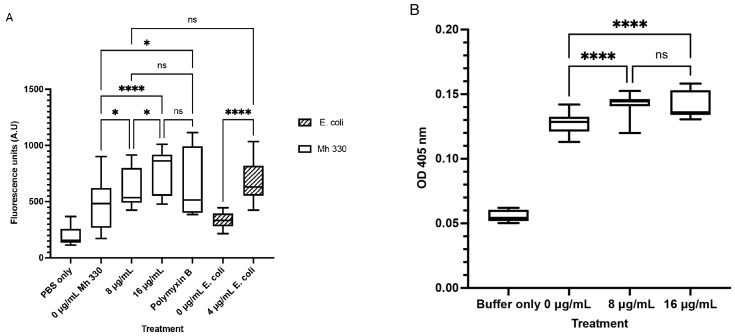
Membrane permeability of WK2 against Mh 330. (**A**) Outer membrane permeabilization characterized by NPN fluorescent dye uptake (* *p* < 0.05, **** *p* < 0.0001, ns: *p* > 0.05). (**B**) Inner membrane permeabilization measured by ONPG cleavage by -galactosidase (**** *p* < 0.0001, ns: *p* > 0.05). Statistical analysis was performed using one-way ANOVA with Tukey’s post hoc test. Results from three biological experiments performed in at least triplicates are presented as box and whisker graphs, where the line plotted in the middle is the median.

**Figure 6 ijms-25-04164-f006:**
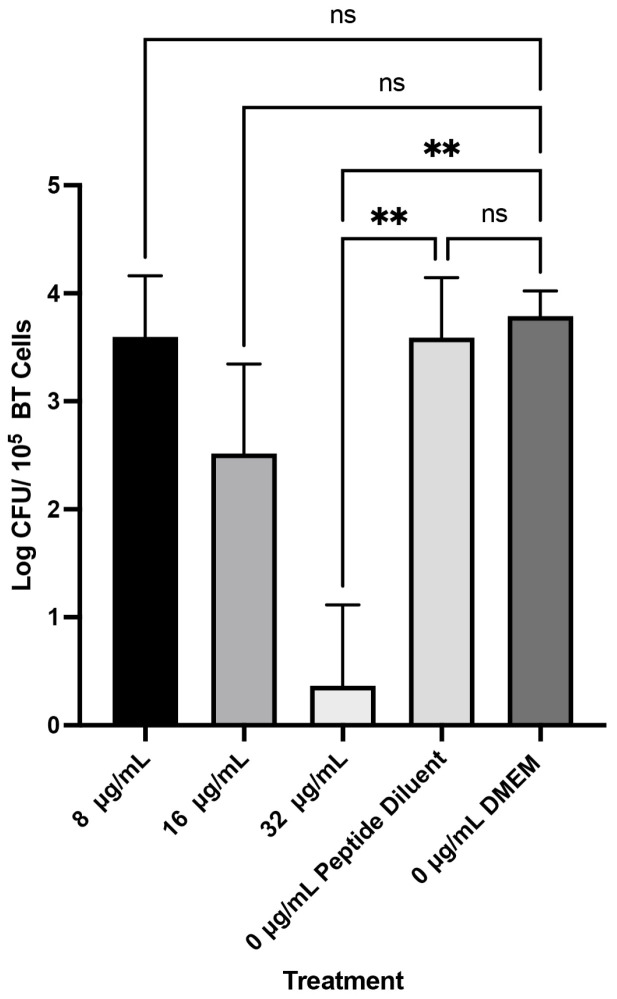
WK2 exerted bactericidal activity against Mh 330 in coculture with bovine turbinates; 32 ug/mL of WK2 reduced *M. haemolytica* bacterial population by approximately 3 logs CFU (** *p* < 0.01, ANOVA Tukey’s post hoc test). No significant differences observed in bacterial population between peptide diluent and DMEM media control (ns: *p* > 0.05, ANOVA Tukey’s post hoc test). Means and standard deviations from results of three biological experiments performed in triplicates are presented as bar graphs.

**Table 1 ijms-25-04164-t001:** Peptides sequences used in this study [17,18,19].

Peptide	Sequence	Structure	Molecular Weight	Net Charge	MIC *E. coli* ATCC 25,922 µg/mL	Parent
PRW4	RFRRLRWKTRWRLKKI-NH2	α-helical	2298.89	+10	9.2	PMAP-36
WRL3	WLRAFRRLVRRLARGLRR-NH2	α-helical	2350.92	+9	4.0	Leucocin A
WK2	WKWKCTKSGCKWKW-NH2	β-sheet	1854.28	+6	4	Leucocin A

**Table 2 ijms-25-04164-t002:** Multidrug-resistant *M. haemolytica* strains belonging to serotype 1 and 2 used in this study.

Strains	Serotype	Sensititre Resistance Profile
Mh 276	1	SDM, SXT, CLI, TYLT, SPE
Mh 330	1	SDM, CLI, TYLT
Mh 13	1	OXY, SDM, SXT, TIL, TUL, CLI, NEO, TYLT
Mh 535	1	Susceptible to all antibiotics tested
Mh 587	2	Susceptible to all antibiotics tested
Mh 136	1	AMP, CTET, FFN, GEN, OXY, SDM, TIL, TUL, CLI, DANO, ENRO, NEO, PEN, TYLT
*E. coli* ATCC25922	O6 biotype 1	N/A

Ampicillin (AMP), clindamycin (CLI), chlortetracycline (CTET), danofloxacin (DANO), enrofloxacin (ENRO), florfenicol (FFN), gentamicin (GEN), neomycin (NEO), oxytetracycline (OXY), penicillin (PEN), sulphadimethoxine (SDM), spectinomycin (SPE), streptomycin (STREP), trimethoprim/sulfamethoxazole (SXT), tetracycline (TET), tiamulin (TIA), tilmicosin (TIL), tulathromycin (TUL), tylosin tartrate (TYLT).

**Table 3 ijms-25-04164-t003:** Minimum inhibitory concentrations for PRW4, WRL3, and WK2 against multidrug-resistant *M. haemolytica* strains.

Strain	PRW4 MIC (µg/mL)	WRL3 MIC (µg/mL)	WRL3 MBC (µg/mL)	WK2 MIC (µg/mL)	WK2 MBC (µg/mL)
Mh 330	>256	16–32	32	8–16	8–16
Mh 587	64	2	16	2–4	4
Mh 276	256	16–32	32	16	16
Mh 136	>256	16	16	32	32
Mh 13	>256	16–32	32	16–32	>128
Mh 535	>256	4–8	8	16	16
*E. coli* ATCC 25922	8	32	32	4	4

Minimum inhibitory concentration (MIC), minimum bactericidal concentration (MBC).

## Data Availability

Dataset available on request from the authors.

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
