# Peer review of "Antimicrobial Activities of α-Helix and β-Sheet Peptides against the Major Bovine Respiratory Disease Agent, Mannheimia haemolytica"

_ijms, 2024, doi:10.3390/ijms25084164_

Round 1

Reviewer 1 Report

Comments and Suggestions for Authors

WK2 and WRL3 synthetic peptides were tested for their antimicrobial activity against M. haemolytica strains. The cytotoxic effect of these peptides on bovine turbinate cells was also evaluated. Additionally, the effect of these peptides on biofilm formation by M. haemolytica was investigated. The manuscript contains significant novel elements and deserves to be published.

Introduction:

This section is well-written and informative. It contains sufficient information, engaging the readers with the problem.

Results

Line 98: It may not be necessary to cite other studies in the results section. Please revise the text by removing the cited references and placing the comparisons in the discussion section.
Table 1: Please provide the units for the MIC value against E. coli.

Table 2: "Susceptible" - to which antibiotics? To all of the tested antibiotics? Please specify.

Discussion

The discussion is very well-written. It reflects the obtained results and provides good explanations for the mechanism of action of the tested peptides.

Materials and Methods

The materials and methods were well-described and allow for the repetition of the experiments.
Line 351: "Wells containing bacteria only and peptide diluent (50 µL of 105 CFU/ml) served as negative controls." - The sentence is confusing. Is the negative control bacteria only, or peptide only? Please revise.

Reviewer 2 Report

Comments and Suggestions for Authors

In the manuscript entitled " Antimicrobial activities of α-helix and β-sheet peptides against the major bovine respiratory disease agent, Mannheimia haemolytica", Bao and co-authors evaluated the inhibitory effects of PRW4, WRL3, and WK2 antimicrobial peptides against multidrug-resistant (MDR) M. haemolytica isolated from Alberta feedlots. The overall theme of the manuscript looks interesting and written well. However, there are some limitations which should be addressed.

The number of replicates in some places is not clear and there is no explanation about the biological and technical replicates. In lines 136, 390, 406, 441, 468 and few other lines, what are your biological and technical replicates? Please explain more about them. For example, in line 136, you mentioned “Three biological experiments performed in at least duplicates are presented as box and whiskers graphs”, meaning that you had at least 6 replicates for 0 ug/ml but what you mean from biological replicate is not clear.

Line 346: “ranging from 0.25 to 128 μg/ml”. Please write the exact concentration of this serial dilution.

Line 352: “The MICs of PRW4, WRL3, WK2 were conducted in at least 2 independent trials with two technical replicates within each of the runs”. Add figures from the MIC assays (Put a picture from the MIC of WK2 plates as one of the main figures and pictures from the MIC of PRW4 and WRL3 plates as supplementary figures and show exactly location of wells for peptide solution at concentrations ranging from 0.25 to 128 μg/ml as well as that of positive nad negative controls).

Line 19: Make M. haemolytica italic

Line 94: Please write minimum inhibitory concentration (MIC)  

Line 120 and Table 3: Please write minimum inhibitory concentration (MIC) and minimum bactericidal concentration (MBC) as footnotes.

Line 375: How many wells were allocated to controls?

Line 378: How many wells were allocated to bovine turbinate cells? Do all these wells with bovine turbinate cells contained peptides? Did you test a single peptide in a single 96-well plate?

Thanks

Round 2

Reviewer 2 Report

Comments and Suggestions for Authors

Dear Authors, 

Thanks for revising your manuscripts.

Two minor comments:

Line 94: The first time you define an abbreviation you need to mention what it does stand for. The correct writing format is "(minimum inhibitory concentration (MIC) 2 to 32 μg/ml)"

Line 500: Delete “Click here to enter text. “

Thanks

Author Response

Thank you, we have corrected as suggested.